# Performing Feces in Contemporary Video and Performance Art in Israel

## Nissim Gal

Art History Department, University of Haifa, Haifa 3498838, Israel; ngal@univ.haifa.ac.il

**Abstract:** In its political ideology, large sectors of Israeli society hold the belief that only people who share its ethnocratic values can share the same hygiene identity with it, reflecting its self-perception as a pure national subject. This is the context in which scatological works based on radical materialism and ethical critique first appeared in Israeli performance and video art at the turn of the twenty-first century. The artworks under discussion seek to consider humankind as machines that produce waste, with an emphasis on the excess waste that separates those who are excluded from the dominant Israeli-nationalist-Zionist view or discourse. Some artists employ excrement as a tool to degrade power structures, while others see it as a source of creativity and an alternative way of material and ethical life. Performing feces, or being shit, constitutes a position of creation, observation, and being to which we should pay particular attention at this moment in time.

**Keywords:** scatology; abject; Israeli art; nationalism; performance and video art

## Fecal Materialism

Through their actual presentation or through their metaphorical or synecdochical representation, the use of feces and other excretions is becoming more prevalent in contemporary Israeli art. This is a result of materialism, which seeks to treat human beings as waste or as machines that produce waste. It also transforms the neoliberal logic of creativity and new life, as embodied, for example, in the national Israeli-Zionist vision, into a logic of decomposition that looks directly at excess and waste as signifiers of those who are expelled from the view or from the social discourse (see Tamsin 2016, p. 115) of that vision. The works discussed in this article find in feces and other excretions evidence of those who have been marked as flesh, body or object, animal or disease, by the major spokespersons of the national Zionist discourse. Excreta and excrement represent a significant contrast to the collectivism known as the Israeli subject, government, and regime.

Waste, feces, or other excretions are at the other end of the spectrum of meanings that Jacques Derrida deals with in his book *Clang: Glas* (Derrida 2021). In her discussion of the writing space in the page design of *Glas*, Gayatri Chakravorty Spivak claims that the graphic architecture in Derrida's book reveals a conflict between two "columns": Hegel and Genet; and, respectively, between mind and body.[1] The artists to be discussed here join the Genetesque column, a material writing of a "peristaltic legend that produces the fecal column" (Gayatri Chakravorty 1977, p. 30). Derrida's reference to the material world and the significance of the issues it raises serves as a springboard for Karen Bray's insightful analysis, in which she argues that dealing with excretion requires examining those whose bodies are seen as surplus or waste because they are perceived as animals, bodies, or other types of objects. In addition to the emphasis on the brain as a metaphor for form, formulation, composition, reason, and consciousness, she suggests in her critical discussion of neo-materialism that one should also consider feces as a significant layer of being. She thus suggests that, in parallel with the metaphor of being as becoming brain, one should look at the metaphor of becoming feces (Bray 2016, p. 108). As a metaphor, the brain represents those in positions of authority, those who generate, produce, and act.

Excrement, in contrast, is linked to those who are seen as expendable, including immigrants, the homeless, women, queers, transgender people, the sick and destitute, and even the environment itself (Bray 2016, p. 111). Shit as a material or as a dominant factor in the discourse on materiality confronts the existing hierarchies in which there is a precedence for power and control. It allows for challenging the hierarchies between body and mind, between organic and inorganic, between the commitment to bringing about transformation-change-"progress" on the one hand, and disintegration, challenging the closure of the self, determinism, and perfection, on the other hand. As Bray contends: "Our materiality and ontology are transformed through what we consume, how we digest it (or fail to do so), and—after it becomes a part of us—how we discard it/us through the joint endeavor of becoming feces" (Bray 2016, pp. 109–10).

The artworks under discussion here will serve as evidence of a process that has been referred to as "performing feces" or as "becoming shit", after Bray. In the present context, it is a call to abandon the racial and gender purity, and the territorialism that goes along with them, as well as the social blindness brought on by Zionist nationalism, in favor of recognizing the significance, existence, right and, moreover, obligation towards those who have undergone an instrumental process of objectification or fecalization. To perform feces or to become shit is to create space for those who are seen as failures and to identify with them, to express profound solidarity with them, to acknowledge the victims of nationalism and heteronormative dominance, the victims of purism and racism in Israeli society, as well as others who have been forcibly expelled from the social body of the Israeli-Zionist national subject.

Scatological aspects and anal gestures were warmly embraced by Israeli art at the turn of the twenty-first century. The artists Zoya Cherkassky, Natali Cohen Vaxberg, Yasmin Wagner, Ariel Bronz, and Roee Rosen have all made it their mission to create anal/scatological art that is critical and unrelenting. The act of emptying the bowels, the "secretion ceremony", and the visibility and centrality of the anal gesture in Israeli art should be understood as a reflection on Israeli society, which constructs for itself an image of a perfect, virtuous, and sacred society through strategies of cleanliness, purification, and symbolic "disinfection" processes that exclude as abject anyone who is not part of the Jewish nationality. The way Israeli society treats anyone it identifies as abject or outcast reflects not only its anxiety about the Other outside, but also reflects a violent doubt operating within the Israeli national body itself. This is a body/subject/nation that strives to escape its responsibility to others, to materiality, historicity, and to an external and ancient residue that nests inside it. The works of art at the center of the discussion in this article support Norbert Elias's finding regarding the development of health and hygiene prohibitions, as being closely related to moral principles, social and political processes, and apprehension about coming into contact with other people (Elias 2000).

Works that deal with the pores of the body and their fluids have been linked to disgust, horror, abomination, terror, fear, phobia, and anxiety, as well as to release, excesses, and happiness. These artworks can evoke a sense of cultural and psychological regression. Excrement, like other substances released by the body, is considered a repulsive, impure substance. The excrement travels from the interior of the body to the outside and undermines the perception of boundaries that give one a sense of security. It attests to the body's frailty and its limitations, as well as to the danger that lurks at the door of the self. "The threat to the boundary or orifice", writes Rina Arya, "is an indispensable part of the modality of being" (Arya 2017, p. 57).

Representations of urine and feces and the use of abject materials and scatological themes already appear in the early history of art, for example, in Roman art in Pompeii. In modern times, one can find such representations too in the works of Rembrandt, Francois Boucher, Paul Gauguin, and others (see Ganim and Persels 2004; Chu 1993; Kuspit 1999, 2008 "In the Anal Universe", and "The Triumph of Shit"). In his *Lectures on Aesthetics*, Hegel contended that Christian and Romantic art turned the abject into a preferred object; the bleeding wounds of Christ embodied the results of human evil, the

tortured body, the marks of the "basest abjection"—all these are the source of the birth of heavenly beauty (Koerner 1977, p. 7). Throughout the twentieth century, and especially in its second half, as well as in the transition to the twenty-first century, scatological motifs became increasingly common in painting, performance art, and new types of media, such as in the works of the Belgian artist Wim Delvoye. According to Joseph Leo Koerner, "Abject Art" took a central place in art at the end of the twentieth century and began to go beyond the use of materials or objects that are perceived as repulsive, or from referring to them, to that of an attack on objects that are perceived as pure, as lacking any affect (Koerner 1977, p. 6).

It appears that the influence of the late 1990s on international art is what has led to the introduction of excrement into current Israeli art. According to Gideon Ofrat, "'pissed art' is the pulsing of an avant-garde that, just before breathing its last and leaving the stage to the postmodern pluralism shuffle, makes radicalization efforts in its final moments." Ofrat asks rhetorically: "How many times will hearing the words "pee" and "poop" make us gasp in shock? How many times will seeing urine and excrement make us gasp in horror? Anyone else bothered to poke a flag through their butt? Who else is enthusiastic about Duchamp's toilet? We've matured, dammit." (Ofrat 2019) Ofrat portrays "abject art" in hues of temporariness and superficiality, as if it were a modern and late manifestation of intra-artistic decadence. The artists discussed below, however, offer distinctive messages to Israeli art, culture, and society, contradicting Ofrat's assertions.

*Poop Instead of Blood*

Natali Cohen Vaxberg posted the video *Vote Glycerin Suppository* on YouTube in 2013, in which she appears as a glycerin suppository in a "propaganda film" for the Israeli parliamentary elections. The video was shot in a bathroom. Cohen Vaxberg's head was emblazoned with a decorative image of a piece of feces, with another similar image on one of her breasts, and a third one replacing the Star of David symbol in the center of the Israeli flag hanging on the bathroom door. The artist speaks in the video work as if she is a glycerin suppository:

> Without me all this shit wouldn't have been here [. . . ] what thrilled the Jewish soul (quoting here the Israeli national anthem, NG) [. . . ] until I came [. . . ] deep deep in her ass . . . on this it was said "Out of the depths I cry out to thee." (quoting here Psalms 130:1, NG) [. . . ] In the upcoming elections, vote for GS— glycerin suppository.

Cohen Vaxberg (2013) formulated in the video *Vote Glycerin Suppository* an artistic manifesto that mocks the existential pathos that characterized the Israeli national (election) discourse, in order to expose the abject in Israeli society. She rejects the state, the nation, and authority. She has stated that the one who "controls, chooses, and enacts ignorant and idiotic laws" is "the one who shits on you from above". She sees voting in elections as complicity with oppression rather than as an act of democracy. She states elsewhere: "Why would you vote in this? Ask your dog or kid to poop on you; I assure you that that stink is superior. Like, if we're talking about feces, do something far more beneficial and do diarrhea on a flag rather than inserting a ballot in the ballot box." (Cohen Vaxberg 2014a) That is exactly what she has achieved in her work *Poop instead of blood*, in which she and the artist Yasmin Wagner are photographed defecating on dozens of flags and pennants, including the Israeli flag, as well as a dog defecating on a map of the world—all to a background of Chopin's music.[2] Cohen Vaxberg explained the reasoning behind the video: "With so much blood, the country and the world soil all these flags. Why is poop scarier than blood? From the severing of heads? From cadavers? What exactly is the taboo? What is it about poop that is forbidden to be discussed?" (Kobo 2016).

When the work *Poop instead of blood* was uploaded to the Internet it caused a scandal, and the rapper known as "The Shadow" even encouraged his Facebook followers to complain to the police about the artist. On 2 November 2014, five police detectives did indeed visit Cohen Vaxberg in the middle of the night, showed her a photo of her defecating

on the Israeli flag, handcuffed her, and drove her to a detention facility where she spent the night. They did this because they believed she had violated the flag law, which states that "offending against the state flag or the state symbol, or besmirching its honor, or using it in a way that is intended to harm its honor, the punishment is imprisonment for up to three years or double the fine [...]".[3] The artist was placed under house arrest after being brought before a judge; her computer, tablet, and mobile phone were confiscated, and she was required to report regularly to the police station and refrain from any network activity for thirty days.[4] (Gaaton 2016) The video, according to the Tel Aviv District Police, constitutes "a blatant and serious act of harming and disrespecting the symbols of the state and the feelings of the public." ("The police arrested an artist who pooped on the Israeli flag.") It wasn't the first time the Israeli flag had been associated with feces[5] but the video *Poop instead of blood* struck a chord. The Israeli public, like the rapper "The Shadow" and his supporters, and certainly like the hundreds of talkbackists who responded to the work's coverage, were outraged, disgusted, and taunted and, in turn, Cohen-Vaxberg's critical enterprise was discredited. The artist was portrayed as a "stupid creature", mentally ill ("Natali's morbid phenomenon is called a mental illness"), someone whose "parents had failed to educate her", and one who will "masturbate with a synagogue pomegranate."[6] In her comments on the work, Cohen-Vaxberg stated:

> The video's statement contradicts every country in which a symbol, a flag, and a religion take precedence over human dignity and freedom. I urge people not to be killed, but rather to refuse and shit on flags. The flags do not cry or suffer. He who died in the name of the flag chose blood over poop and war over love. Whoever killed in the name of the flag harmed an entire family, which is a little more serious than spitting on a flag and inflicting emotional harm on the public. What happened after I posted my video demonstrates what happens when these symbols are taken too seriously. I was arrested and violently questioned. Hitting a rag, on the other hand, is not considered violent. On the contrary, it can prevent violence because the man is able to express his rage. If people are not allowed to protest, if they are denied all forms of expression that should be available in a state that calls itself democracy, they may resort to physical violence (Ehrlich 2016).

The act of defecating on a national flag embodies symbolic violence and anti-nationalist criticism that undermines the established order's sanctity and integrity. Cohen Vaxberg and Wagner are not alone in this respect; there are numerous examples of people defecating on flags, with, as Nadav Neuman wrote, the majority of those defecating on flags being women, and the majority of those shocked being men. A woman who shits on a national symbol is considered a taboo breaker and a threat to the entire social order (Neuman 2016).[7] Cohen Vaxberg's works echo the radicalism of "Vienna Actionism" (*Wiener Aktionismus*), which presented performances in the 1960s that included actions involving bodily waste discharge in a public space in general, and in the university space in particular, as well as artistic depictions of sex and violence. Members of the "Viennese Actionism", which included coprophilia as well as eating excrement, filmed and distributed video clips, and turned these actions into cinematic works. These works are not those of scatological pornography; rather, they seek to expose the darkness in the human soul and society that were involved in the acts of Nazism (Verrips 2017, p. 31). Cohen Vaxberg's work thereby contributes to an artistic tradition of criticizing nationalism through the use of abject materials. Her work is also part of a feminist critical artistic lineage: *Poop instead of blood* is a companion piece to Kiki Smith's *Tale* (1992), which featured a naked woman kneeling on all fours with a long tail of feces dangling behind her (some say the umbilical cord, but in any case material identified as abject), which locates her on the threshold between the human and the animal. Cohen Vaxberg and Wagner share Smith's rejection of sterile purity embodied in the conflict between animal and human, nature and culture, dirt and gender, Freudian psychoanalytic paternalism and Darwinian evolutionism (Kauffman 1998, pp. 261–62).

At the level of mental economy, Cohen Vaxberg's work challenges the notion that "those who do not control body closures" should experience shame. The use of feces in the video is seen by the public as evidence of the infantility, stupidity, animality, and failure of the two artists, who did not properly complete the anal stage, and thereby remind us of our potential for failure, since we too are prone to infantilization.[8] Moreover, Cohen Vaxberg perceives her actions and their results as a graphic expression of contempt: "States and governments always shit on the citizens, once we shitted on the governments." (Gillerman 2016).

According to Mary Douglas's book *Purity and Danger: An Analysis of Concepts of Pollution and Taboo*, certain persons or items are perceived as being dirty or polluted due to cultural factors. She contends that the social system produces a distinct hierarchy and order of categories that are employed to maintain the boundaries between entities—boundaries that when crossed result in contamination. The dietary laws in Judaism are one of the most blatant examples of such restrictions. Food, for instance, is appropriate during a meal but is deemed pollutant when it stains one's shirt. It is possible to identify in Douglas's explanation a foundation for the violent emotions sparked by Cohen Vaxberg's sensational work as one that violates social norms and regulations, contaminates them, and creates impurity. Douglas engages extensively with the concept of holiness in Judaism.

> When rituals express anxiety about the body's orifices the sociological counterpart of this anxiety is a care to protect the political and cultural unity of a minority group. The Israelites were always in their history a hard-pressed minority. In their beliefs all the bodily issues were polluting, blood, pus, excreta, semen, etc. The threatened boundaries of their body politic would be well mirrored in their care for the integrity, unity and purity of the physical body. (Douglas [1966] 2001, p. 125)

In the third chapter, entitled "The Abominations of Leviticus", Douglas claims that "the Hebrew root of k-d-sh, which is usually translated as Holy, is based on the idea of separation." (Douglas [1966] 2001, p. 8). She adds that "the next idea that emerges is of the Holy as wholeness and completeness" (Douglas [1966] 2001, p. 52). The commandment to be a holy nation, or sacred, is closely related to the laws of kashrut. Impurity is a deviation from the commandment and the order: "All bodily discharges are defiling and disqualify from approach to the temple" (Douglas [1966] 2001, p. 52), and the demand for physical perfection appears "in the social sphere and particularly in the warriors' camp" (Douglas [1966] 2001, p. 52). The idea of perfection is also at the basis of the ban on hybrids ("Hybrids and other confusions are abominated") (Douglas [1966] 2001, p. 54), as well as at the basis of "rules of sexual morality [that] exemplify the holy" (Douglas [1966] 2001, p. 54), and the laws of kashrut, which are a development of the concept of order and sanctity.

This puritanical idea that separates the sacred, whole, and undamaged from the filthy, impure, partial, and damaged is challenged by Cohen Vaxberg's work in general, and *Poop instead of blood* in particular, and the divide is trampled on harshly. She is not content, however, with merely dumping the dirt in the improper or inappropriate location; rather, she also uses biological matter and is, therefore, contaminating. The use of bodily secretions is what has set Cohen Vaxberg's protest action apart from other protests against the Israeli flag, such as those by some of the radical ultra-Orthodox factions, which have met with less resistance and which have not always resulted in the intervention of State institutions. This intensified sense of defilement and disgust is what sparked the fierce attack against Cohen Vaxberg. In her 2014 film *Response to Arrest (The Poop Affair)*, Cohen Vaxberg addressed the Israeli public and the authorities following her arrest by the police. In her speech, she apologizes in a parodic way for the poop, with exaggerated sorrow, while portraying herself as a prisoner being executed by ISIS.

The word "poop", as in the piece *Poop instead of blood*, was chosen by Cohen Vaxberg to be employed in this work's title, rather than "shit" or "feces", which lends the work a satirical tone. Poop is typically used to refer to children's secretions and to emulate the vocabulary of a youngster who wants to play with his secretions, rebelling against his

environment and protesting the restrictions on his behavior before he undergoes a process of "normalization". In Cohen Vaxberg's art, poop is transformed into a disgusting and "hazardous" substance in the eyes of Israeli culture. The talking excrement video brings to mind other scatological incidents in which prison inmates smeared themselves with it or used it as a form of protest, such as the "dirty protest" that took place in 1978 in prisons in Northern Ireland, in which the prisoners refused to leave their cells to shower or use the toilets and instead urinated through the door openings and smeared their excrement on the walls of their cells; or the Guantanamo protest in which prisoners flung feces, urine, and vomit at the American guards. In a reverse action in the Abu Ghraib jail, the guards told the inmates to cover themselves in their own excrement, which was thus utilized to abuse the detainees. In her work, Cohen Vaxberg has been establishing a path to scatological art that engages with Israeli society, and the policing of its symbolic order, by smearing her feces on the Israeli nationalist emblem: the flag.[9]

Children's, prison inmates', and artists' scatological protest acts would not have been successful if the sense of social rejection in the presence of disgusting items, notably excrement, had not existed a priori. According to Aurel Kolnai, who studies the phenomenon of disgust, things that are considered repulsive and filthy are directly tied to the organic and biological, or to what are viewed as byproducts of life. Excrement and sweat, for example, are considered to be intimately associated with dirt (Kolnai 2004, pp. 55–56; Arya 2017, p. 54). Using the term "abjection", Julia Kristeva has sought to explain the aversion to the abject, the sociological and psychological division of the world of materials, into proper and acceptable materials—and abject and despised materials. According to Kristeva, the substances emitted from the body's exit "doors"—saliva, blood, milk, urine, feces, tears—are contaminated not because they pose a threat to the subject, but because they reflect the danger of the symbolic order collapsing (Kristeva 1982, p. 70). Following Kristeva, Cohen Vaxberg places disgust-arousing material at the heart of her works, and the feeling of rejection is a central component of the viewer's experience. In her works, the object of contempt is usually the State. Thus, in her video *The Sudanese departure of Meira Lapidot* (2013b), Cohen Vaxberg criticizes the State of Israel's attitude toward African refugees; in the video *How would you manage without the Holocaust?* (2012), she personifies the Holocaust and accuses the State of Israel of instrumentalizing and exploiting it for its own purposes. "Hello! I'm the… Shoah! [. . .] and I'm the best thing that happened to you [. . .] thanks to me you have a nuclear bomb without being told anything! [. . .] I got used to being your *sharmouta* [slut], but there are also *sharmoutas* in the IDF that don't work for free! I shit on you!" She also insults the country and its flag in *Poop instead of blood* (along with insulting other countries and other flags). The rejection of the State shapes Cohen Vaxberg's independent artistic position. She distinguishes and defines her borders by placing the State as the real abjection, with excrement being the material through which she marks the State as the loathed item, the waste, the dung, the defilement.[10] On the one hand, she is constantly engaging with the country, representing and directing it in her works; on the other hand, she withdraws from it, repeatedly marking her separation and differentiation from it. She is both fascinated and repulsed by the State.

The core of the psychoanalytic story, according to Kristeva, is the abject. The crucial aspect of the abject is reflected during the separation from the mother, prior to the stage where autonomous subjectivity is distilled. Abjection is the process by which the subject consolidates its boundaries through the rejection of objects it does not want near it. The abjection and rejection are a formative experience that affects the reluctance that the subject will experience in their adult life.[11] The abjection process occurs during the pre-Oedipal relationship stage between the infant and the mother (or the figure representing her), at the stage when the infant perceives the mother's body as an abject. Abjection is thus required for separation from the mother: on the one hand, the encounter and union with the mother during the breastfeeding process, and on the other hand, the withdrawal and separation intended to mark a gap and difference from her. There is both peace and a sense of merging of mother/child in this process, but also terror and disgust. And the baby must

eventually abject the mother's body (if she does not respond, if she does not satisfy him) in order to avoid being abjected itself. This is the foundation for the emergence of the child's autonomy, which includes a series of cleansing rituals, body manipulation, and body care in the subsequent development process.

Unlike Kristeva's narrative, Cohen Vaxberg's act of protest is not directed at the primary, primordial, fundamental mother, who exists outside of time and space. She makes no reference to the woman's pre-discursive or extra-discursive status. Her criticism is directed against the law, or in front of it, rather than beyond it. The abjection in her work clashes with culture, such as when she declares, for example, her refusal to be a mother and fulfill her "duty" according to the patriarchal law. In the spirit of Judith Butler's words, who criticizes Kristeva for "keeping the concept of culture as a paternal structure and delimiting motherhood as a reality that is essentially pre-cultural", and for the naturalizations of the maternal body in her writing "effectively objectify motherhood and prevent an analysis of its cultural construction and of its difference" (Butler 1990, p. 80), it can be said that Cohen Vaxberg's scatological art confronts what is perceived in her eyes as an action, a law, or a nationalistic paternalistic framework. Cohen Vaxberg believes that the female body operates within the law and should not be embodied outside of it. Butler asserts that, if critical activity is possible, then its field of action must be within the framework of the law, and it exists through the application of the law against itself; liberation cannot exist through a naturalization that evokes a "natural" past, a primal pre-cultural pleasure, but, rather, through action whose purpose is to break the law in the present (Butler 1990, p. 93). The maternal body serves as the reference object for Kristeva's theory of the abject in both the phantasmatic and material senses. However, this theory can also be viewed as a structural idea that depicts the abject as something that disturbs limits, order, system, and identity (Steihaug 1998, p. 36).

*Love the Juice*

Ariel Bronz presented the performance *Love the* Juice (2015) in 2016 at the "Israel Culture Conference", held by *Haaretz* newspaper, which addressed freedom of expression and was themed "Culture Demands Independence". Bronz's performance featured a character who harbors racist, homophobic, and Zionist sentiments.[12] At the conclusion of the performance, the actor was booed; in response, he threw oranges, which had been used as props in the performance, at the audience members who had booed him. Some of the audience then started to leave but the disturbance persisted. Following the shut-down of the microphone by the event's organizers, Bronz yelled at the audience, "Censorship", and exited the stage into the hall while carrying a white flag with a dashed outline of the Israeli flag/emblem, resembling a flag for coloring-in. The event moderator thanked him and asked to get on with the conference program, but Bronz ignored her instructions and, returning to the platform, he smacked his butt with his hand, much to the moderator's displeasure. She turned to him and asked, "Why the provocation?".

A spectator then yelled "Come on, throw him off the stage!" Bronz stuffed the flag into his buttocks and continued to walk across the stage in this way until the security personnel forcibly removed him. It appears that it was the flag's placement in the anus that incited the crowd's rage. Culture Minister Miri Regev, who was present in the hall, was quick to respond in her demand that a police investigation be opened against the artist:

> Tucking the Israeli flag up one's buttocks is not art; it is a desecration of the flag's name, for which soldiers and civilians in the State of Israel are killed. It's not only not funny, but it's also suffocating. When I said today that I will protect our culture and will not allow destruction and devastation to occur under the guise of free expression, I was met with boos. So here we have the same people who yelled contempt for the country watching this disgusting show. Is this culture? Are these cultured people? Do you want me to pay for it? Certainly not. (Walla Culture 2016; Cohen 2016)

Bronz, in contrast, explained the anal gesture as one of critical linguistic expression:

I come from Odessa, Ukraine, where there is a very well-known metaphor that is common in many dark regimes, which in direct translation says: "Flag in your ass". Have a flag in your ass—it's a wish you make to people who aren't ready to listen or hear. My act was done in the heat of the moment, in response to the audience's enthusiasm and desire to bite into my flesh, but I believe it was done on the basis of the metaphor I grew up with. Let whoever can't listen have a flag in his ass (Biron 2016).

The question is why the act of sticking the flag in one's behind so shocked the audience and the culture minister—that same minister whose speech at the same conference began with the phrase "cut the bullshit", which clearly has scatological resonance. It seems that Bronz's action, which outraged many, should not be seen only against the background of this specific critical action, but also against the background of the content of the show itself, during which Bronz—and the character he plays—says that he and his "Palestinian friend", "we'll call him Jafa", "we decide to enter the orchard for a quickie". He continues with the plot: "while we lie drunk"—at this point in the performance, Bronz is lying on his side on a piece of cloth with a map of Palestine printed on it and his legs are spread in the air, between two phallic orange trees, as if to mark the movement of the bodies and the anal penetration that took place during that "quicky", and he tells the audience about an incident he defines as a "miracle": "I didn't control my hands, I didn't control my body wastes", and as a result he violently attacks the friend: "You don't know what it's like to urinate on a bruised body of a smelly Arabush [a derogatory term for an Arab in the Hebrew slang NG]... The terrorist fled from me into the orchard and left me sprawled in a huge puddle of urine, blood and dirty Arab semen".

The reaction of the audience and the minister, as well as some of the journalists who wrote about the event that led to the artist's interrogation by the police, can be seen as an expression of horror and rejection that resulted from the lost distinction between a complete, satisfied, just, and moral self, inherent in the perception of Israel as a pure national subject, as an imaginary whole, and the debased image or object, which Bronz placed before that self as a mirror. Bronz's narrative and performance, which included depictions of violent relationships, trauma, foul odors, blood, urine, and sperm, as well as the gesture of the anal insertion of a flag, elicited outrage and disgust from the audience because these gestures were perceived as gestures of contempt associated with insult and immorality.

Kristava quotes the prophet Ezekiel (4:12) in her writings on abjection: "And thou shalt eat it as barley cakes, and thou shalt bake it with dung that cometh out of man, in their sight" (Kristeva 1982, pp. 108–9). The barley cake offered to man is actually animal food; man has become like an animal, and his excrement represents his impurity, the sin he committed. "A mouth attributed to the anus: is that not the ensign of a body to be fought against, taken in by its insides, thus refusing to meet the Other?", asks Kristeva. (Kristeva 1982, p. 109) In Bronz's performance, the violence, the urine, the sperm, and the allusion to the corpse are markers of the debased and polluted that must be kept away or repressed by the subject, because they stain the private and national body with impurity, with abomination. Bronz's action crosses the most fundamental lines between the spiritual, symbolic, cultural, and the animal and beastly. The anus is a passageway for feces to exit the body; but, in Bronz's show, the action is the inverse. The anal gesture contradicts masculinity and even subjectivity; it violates a prohibition that is "required" for the establishment of sound subjectivity. Bronz returns to the present's pre-cultural stages, transforming himself into a jumble of body parts; mouth, eyes, and anus. He marks a retreat to a pre-symbolic stage and challenges the socio-political order by engaging with forbidden or despised substances, such as excrement, blood, and sperm, as well as inserting the flag into his own anus.[13]

**Praising Abjection**

Taking pleasure in abjection, the artist Roee Rosen created the video work operetta *The Dust Channel* (2016) for *Documenta* 14, which was held in Athens and Kassel in 2016.

This operetta vehemently criticized the attitude of Israeli society towards immigrants and asylum seekers. Its music was written by Igor Krutogolov, and the libretto, written in Russian by an imaginary character (perhaps Rosen himself), tells the story of a wealthy Israeli couple's love for their Dyson DC07, a British-made vacuum cleaner. The couple, played by Israeli Opera singers Inbar Livne Bar-On and Yoav Weiss, live a typical life that includes eating, sleeping, having sex, and having fun while simultaneously singing a song of adoration for the vacuum cleaner. The musicians who work as the couple's housekeepers exhibit the western fixation on sterile environments, spotless goods, and "pure" races. The cleanliness and purity of the house are under the supervision of law enforcement officials who visit the house. The couple's extreme aversion to dirt and their uncontrollable lust for the vacuum cleaner, whose transparent body and cutting-edge design reveal the dirt that has been removed from the world, are embodied by the extreme maintenance of the house, the servants who live in it, and the law enforcement officials who visit it.

We can see a desert through the window of the house, which appears to be the final destination of *The Dust Channel*—the location to which the couple wishes to transfer the dust. The window delineates the boundary between the house, "our" territory, and the dirt thrown out from it. The bourgeois couple's obsession with cleanliness, dust, and dirt is, at least in the Israeli context, a reflection of the desire for a "home" territory, a desire mixed with xenophobia. "Fear", writes Kristeva, "cements his compound, conjoined to another world, thrown up, driven out, forfeited." (Kristeva 1982, p. 6) The Negev, the desert visible through the window, is where the couple wants to rid themselves of the dirt, sand, and dust. This desert is where the "Holot" was located: an open-air detention facility where Israel housed refugees and migrants, mostly from Eritrea and Sudan. The facility was established and used as a space that contained the "sands" that Israelis seek to repress and contain. The refugees were housed like dirt in the desert prison, removed first beyond the social border and then beyond the national territorial border. Beyond the border, the refugees are what is excluded; they are the Other.

When the couple goes to bed at night, the vacuum cleaner comes to life. It enters the living room and fumbles with the TV remote control, watching familiar vacuum cleaner advertisements and pornographic clips that feature vacuums on the various channels. The video work occasionally also incorporates quotes from Israeli political figures. Cleaning supplies and Israeli politicians appearing on television share the desire of the Israeli couple to cleanse society of dirt, disease carriers, and criminals. As is well known, certain Israeli politicians have made little effort to hide their resistance to the migrants, considering them a threat and "filth": Danny Danon, a member of the Knesset, stated "People here are angry because they believe their communities have turned into refugee camps. [. . . ] Expulsion of the intruders from the State of Israel is necessary. We must remember the rights of Jews who reside in the State of Israel" (Shalev and Shimoni 2012). Miri Regev, Knesset member at the time, stated that: "The Sudanese are the cancer in our body [. . . ] We will protect our children, our women, until the last of the Sudanese leaves" (Shalev and Shimoni 2012), and the Interior Minister Eli Yeshi claimed that the rise in crime and the spread of terrorism was caused by refugees, or "infiltrators" in his terminology (Ephraim 2012). What is evident from Danon's warning to not "neglect the rights of the Jews" is that the filth that the bourgeois household is striving to purge itself of in Rosen's operetta is that of the impurity that threatens the Jewish *ethnicity*. According to Kristeva, cleanliness, similar to purification rites, serves to keep undesirables, such as the unclean and impure, at a distance. This is because "the logic of avoiding filth . . . founded the 'self and clean' of each social group, if not of each subject" (Kristeva 1982, p. 65).

Israeli fondamentaliste politics associates migrant workers and refugees with the abject and the debased. According to Kristeva, the abject is not an object, but it does have one distinguishing feature: it is opposed to the ego and is "the jettisoned object" that draws us to the place where meaning collapses. Kristeva contends: "A certain 'ego' that merged with its master, a superego, has flatly driven it [the abject NG] away. It lies outside, beyond the set . . . And yet, from its place of banishment, the abject does not

cease challenging its master" ([Kristeva 1982](), pp. 1–2). Because the abject does not respect established borders, positions, or rules, it poses a threat not because it is dirty or ill in and of itself, but, rather, because of its critical potential and ability to upend identity and order. When Israeli politicians employ the metaphors of illness, filth, and risk to women, children, and Israeli society at large when discussing refugees, they are implying that the destitute or the abject could pollute the Jews. The desire for a clean body shared by the affluent couple in the operetta, and by Israeli culture as a whole, is irreconcilable with the presence of bodily secretions or dirt. The secretions blur the boundaries of the body, the signs of autonomous and pure selfhood, and make them ambiguous and debatable by challenging the separation that might be self-evident between the self and the other, between the inside and the outside, between "my" subjectivity and that of the refugee.

Abjection is a process, an action, or a cultural mechanism that expels certain groups by tarnishing or portraying them as contaminated. Xenophobia is fueled by abjection, with not only foreigners but also marginalized groups perceived by society as ugly, dirty, and capable of instilling fear or nausea. Abjection also encompasses homophobia, racism, sexism, and ageism. To maintain the boundaries of oneself and the difference between oneself and the Other, the subject reacts to the abject with rejection, abhorrence, and disgust. The alienation and denigration of the Other is regarded as a necessary condition for the establishment of the subject or of society ([Alphen 2017]()).

The expulsion of the Other is thus required in order for the subject to establish itself. In Butler's words: "The subject is constituted through the force of exclusion and abjection, one which produces a constitutive outside to the subject, an abjected outside, which is, after all, 'inside' the subject as its own founding repudiation" ([Butler 1993](), *Bodies That Matter*, p. 3). Abjection is used in social and cultural life to define selfhood, to exclude things that threaten the systems and orders that define identity, and to safeguard personal and societal borders, as well as gender, class, and identity divisions. The problem emerges when one recognizes that exactly what we abject, what we place within the variable of abjection, whatever it may be, will never remain outside the limits; despite our best efforts to rid ourselves of it, it will always be inside us and continue to haunt us. We are continually in contact with the "threat" that the abject poses.

There is, therefore, a particular perversion nestled inside the bourgeois home. The deviation and the crossing of borders occur in the center of the bourgeois scene rather than in a distant exotic location, as in the surrealist works that Rosen's work alludes to, particularly the film *Un Chien Andalou* by Luis Bunuel and Salvador Dali.

In one of the scenes from *The Dust Channel*, a woman kneels in front of what appears to be a bodily discharge, and while she is drawn to it like a magnet and even eats it, she simultaneously recoils from it. Here, bodily fluids and secretions are primarily associated with images of women. Hairs are already emerging from her armpits in the opening shot, going beyond what is considered "acceptable" for a lady. She constantly cleans and purifies the house while also devoting herself to bodily secretions, becoming a subversive trope of (feminine) liberation, offering an alternative, disruptive image that rejects the symbolic order. She rejects the oppressive demonstrations of the categories of the beautiful, pure, clean, and national when she is licking the discharge. The woman—and, by extension, Rosen's art—confronts the despised abject; she not only judges but also celebrates it, challenging the sharp distinction between the pure and the pure impossibility, between what is moral and what is immoral. She breaks the rules and sins, inviting the audience to do the same.

The bourgeois household operates in the video as a system of contamination and purification. The secretions that appear around the house serve as a sort of counterpoint to the obsession with cleanliness. A person's way of dealing with what threatens them, with what appears to have its source outside or inside some excessive interior, is cleanliness; the fight against debased abjection. The threatening thing can be imagined but not comprehended; it is both fascinating and disturbing. Rosen "activates" the shit as a metaphor, residue, image, and symbolic material; not only as a source of contempt (abjection), but also as

a beautifying substance, deterring but also tempting, attractive, and repulsive. As such, it echoes the double meaning of the Hebrew word "mag'il", which denotes not only the disgusting, obscene, repulsive, and abominable, but also the act of rendering dishes kosher by immersing them in boiling water, for example, in preparation for Passover, which makes things "proper" according to Halacha (religious law).

Another scene shows the man in his underwear, his hairy legs echoing the woman's hairy armpits at the start of the film. When he bends under the table to clean, male hands are seen massaging his buttocks and simulating an act of intercourse. The position of kneeling on all fours echoes the anal eroticism that pervades Rosen's work and connects the body and desire to animal practices and smells. Sigmund Freud writes:

> Here upbringing insists with special energy on hastening the course of development which lies ahead, and which should make the excreta worthless, disgusting, abhorrent and abominable. Such a reversal of values would scarcely be possible if the substances that are expelled from the body were not doomed by their strong smells to share the fate which overtook olfactory stimuli after man adopted the erect posture. Anal erotism, therefore, succumbs in the first instance to the organic repression which paved the way to civilization (Freud 1930, pp. 98–99).

In other words, the value of olfactory stimuli decreased as mankind stood up and walked erect, becoming tired of the pungent odors emitted by the body; and thus the popularity of anal eroticism declined on the way to the birth of culture.

A halved lemon is seen in the film immediately after the anal scene, with a finger pressing on it in the center, as if penetrating it. The image sequence replicates the Bataille sequence of an eye, a hole, a mouth, and other bodily orifices, as well as secretions, and, as previously mentioned, excrement. The excrement, and by extension all decay and pollution, are evidence of something that is beyond identity, or, as Kristeva puts it, "the ego threatened by the non-ego, society threatened by its outside, life threatened by death" (Kristeva 1982, p. 71).

In Kristeva's mental narrative, excrement is associated with the danger that threatens the subject and his identity from the outside, whether in the representation of the dangerous Other or in the representation of finitude and death. According to Bataille, however, the debased and abject have a liberating potential: "Shit becomes not a symbol of potential shame or disgust but a symbol of explosive protest against rationalism and idealism" (LaCom 2007). Reading the writings of the Marquis de Sade led Bataille to distinguish between "Two polarized human impulses: excretion and appropriation", and he explains: "The process of appropriation is thus characterized by a homogeneity . . . of the author of the appropriation, and of objects . . . whereas excretion presents itself as the result of a heterogeneity, and can move in the direction of an ever greater heterogeneity, liberating impulses whose ambivalence is more and more pronounced" (Bataille 1985, p. 95). According to Benjamin Noys, for Bataille excrement is not about removing, rejecting, splitting from, or getting rid of a foreign body, but about dealing with its heterogeneity; the discharge does not provide controlled pleasure. In de Sade's writings, Bataille finds the idea of secretion moving within a heterogeneous process; what is secreted is not removed to "another place", but is released and moves in an uncontrollable process (Noys 2000; Georges Bataille 1985, p. 125). Rosen's work, like that of Bataille, seeks to liberate heterogeneous impulses from the homogeneity and appropriation imposed on them. The importance of the heterogeneous lies in avoiding world consumption, appropriation, and the erasure of the world's plurality of identities. Rosen's scatological work aims to liberate the heterogeneous, to disrupt the routine, to question the homogeneity of the social order, and to reflect non-normative situations; disruption as a source of freedom.

### "Anus Mundi"

Scatological artworks interrogate situations of transgression and question the definitions of cleanliness and the propriety of bodily functions and materials, regardless of the fact that these are being perceived as impure and unworthy of display or public debate.[14]

The outrage that Cohen Vaxberg's works sparked, among both the general public and politicians, demonstrates their intention to undermine the social order. Cohen Vaxberg and Bronz were both arrested and interrogated by the police, and scorn and anger were directed at her on social media, in press articles, and in the politicians' reactions. As noted following Bronz's performance, his "humiliation" of the symbols of the government through anal gestures led the Minister of Culture to threaten to cancel some of the cultural budget. Such a threat is not uncommon. The use of similar gestures also raised objections to Chris Ofili, whose painting *The Holy Virgin Mary* (1996) presented an image of the Virgin alongside elephant dung and cutouts of pornographic photographs. In 1999, when the painting was displayed at the Museum of Modern Art in Brooklyn, New York, as part of an exhibition of British art—*Sensation*—Rudolph Giuliani, the then mayor of New York, threatened to withdraw seven million dollars in support of the museum until what he perceived as sacrilege was removed. In the end, however, he was forced to back down by order of the court, which did not see the exhibition as a violation of freedom of expression. Bronz's anal gesture recalls Robert Mapplethorpe's photograph *Self-Portrait with a Whip* (1978), in which he inserts a whip into his behind. The photograph appeared in 1990 in a traveling retrospective exhibition called *The Perfect Moment*, which sparked riots, censorship, and boycotts in the late 1980s and early 1990s. In Cincinnati, for example, the owner of the gallery at the Center for Contemporary Art where the exhibition was held was accused (and later acquitted) of soliciting immoral activity and was the subject of a police investigation that included documenting and gathering evidence of "disgusting material" from the exhibition (Palmer 2015).

Israeli society is characterized by a belligerent and anti-critical trend that extends beyond art and is a result of the political–cultural discourse that has consistently controlled the country. This process began in Israeli society in the 1980s with the shift from a social democratic state to a neoliberal capitalist state under the influence of a neoconservative economic social ideology developed in the United States. This process was accompanied by a hawkish foreign policy that showed no tolerance for political pluralism and promoted a nationalist concept that made the occupation of the Palestinian territories "kosher". In an effort to create a homogenous and racially exclusive nation state, global neoliberalism has been translated in the State of Israel into a *völkisch* notion that prioritizes the rights of Jews over those of other citizens and residents. Right-wing extremists have attempted to transform the State's power structure from one based on universal and secular law to one based on racial and religious legislation (Friedland and Hecht 1998). Loyalty laws to the Jewish state have been promoted to that end. The "New Right" party distributed a video in the run-up to the April 2019 elections, in which the Minister of Justice, Ayelet Shaked, appears as a presenter for the "Perfume of Fascism", responding at the end to criticism of her worldview and stating: "To me, it smells like democracy."[15]

The background for the flourishing of scatological works in Israel is socio-political. The 2000s saw an increase in ultra-nationalist trends among certain sectors of Israeli society that were hostile to democracy and pluralism, favoring xenophobia, ethnic discrimination, and social Darwinism, opposing territorial compromises and displaying extreme ethnocentrism and hostility towards minorities, particularly the Palestinian-Arab citizens of Israel (Harel-Shalev and Chen 2015).

Many Israeli artists have been operating in an oppositional situation during recent years, with their work perceived by the government as detached, representing the socio-geopolitical elite located in the center of the country. The conservative line taken by right-wing governments, particularly since the formation of the 34th government, in which Miri Regev served as Minister of Culture, denounced anything perceived as deviating from the national, collective, and conservative. The Law of Loyalty in Culture, which the Minister of Culture sought to enact, is a clear example, as it allows her, or anyone who succeeds her, to reduce or exclude budgets from cultural institutions based on criteria solely determined by the government, and allowing for political interference in these institutions' decisions.[16]

Government-supported budgets, which are the lifeblood of the arts industry, and anti-democratic legislative efforts to advance political and cultural agendas and control culture, are not exclusive to the State of Israel. Steven C. Dubin discusses how any government may employ debates over art in order to obscure significant social and political issues.

> . . . whenever a society is overwhelmed by problems and its sense of national identity is shaky or diffuse, a probable response is for states to attempt to exercise control by regulating symbolic expression. Whether the target is art or language, governments try to demonstrate their continued efficacy by initiating diversionary conflicts . . . temporarily deflect attention away from other concerns . . . Distractions of this sort belie desperation, but they are strategically important. And they frequently center on the arts, a relatively defenseless sphere of activity (Dubin 1992, p. 19).

Cohen Vaxberg made the video *Response to Arrest (The Poop Affair)* after her arrest by the police, following her debasement of the national flag in her previous video work *Poop instead of blood*. The "poop" is used in this later video as a kind of subject that absorbs everything that conservatism condemns. The woman sentenced to death by those who are seen in the eyes of the artist as equivalents of the executioners in ISIS delivers her "last words" in this video. The excrement is given a complex description in her speech/last words: as both abject and rejected, as well as endowed with "magical" and seductive properties. Her monologue is based on the repetitive rhetorical structure of an apology and depicts a series of oppressed peoples who are victims of Israeli nationalism.

> Sorry that my poop isn't blue and white (the colors of the Israeli flag NG), blue and circumcised, rounder, less triangular, not a beadle, doesn't dream of the temple, and knows no commandment. He is blind to flags and poop symbols, skeletons not buried in poop, severed heads, he does not wrap coffins, the entire world is a lavatory for him. Sorry that he can't salute with at least one hand, that he can't vote, that he can't intercept missiles, that he can't shout, death to leftists, lesbians, gays, Arabs, Eritreans, Sudanese, disabled people, dodgers, a-sexuals, women who don't want to have children, and anyone who has an opposing opinion (. . . ) Please excuse my inability to stand when the siren sounds (. . . ) Forgive me in my own name and in the name of my poop, who sinned with pride and pretended to stink like the country. To stink of genius, like capital and power[17] (. . . )

Cohen Vaxberg's satirical excretory apologia mocks the strength of the objections evoked by her work, and it begs the question of why scatological works elicit such horror. Her works have elicited public opposition, among other things, due to the close relationship between the abject and the self: when the abject crosses the boundaries of the self, it acts as a threat and challenges the socio-cultural being. The abject is a "fantasmatic substance" that is in an intimate relationship with the subject, and, as Hal Foster has stated: "this over proximity produces panic in the subject. In this way the abject touches on the fragility of our boundaries" (Foster 1996, p. 153). In addition to her interest in abject materials, images, metaphors, and gestures, the artist's preoccupation with all of these seeks to "expose the mechanism whereby some subjects are expelled in order to objectify the sovereignty of others" (Koerner 1977, p. 5). The asylum seekers in the "Holot" facility are only briefly shown in Rosen's *The Dust Channel*, which does not seek to film them, but rather, to expose the inclusive–repressive logic behind the attitude toward the refugees, in the same context of the I/Other-assigned relationship: "Even when they aren't apparent, the others", according to Rosen, "become the ones who create your identity. Both they and you define each other. It is artificial to erase and "clean" them." (cf. Barzilai 2018).

The scatological works discussed here reveal the frailty of the thin membrane on which our "civilization" stands, as well as the dark side of the culture in which we live. In her work *Vote Glycerin Suppository*, Cohen Vaxberg presents herself as a glycerin suppository that releases the nested violence concealed within the national body. Bronz confronts the

viewers—artists, politicians, citizens, and residents—with the violent meaning of the anal world. "Anus mundi", the anus of the world, is the most filthy and detestable place on earth; it is the filth that justifies the violence against it. The disgust that Bronz's work evokes reflects not only the abhorrence of the anal gesture but also the desire to purify the world.[18] Scatological works can be seen in relation to the words of James George Frazer, who warns us in his book *The Golden Bough* about a layer of wild barbarism that lurks beneath the surface of society and may one day crack it. Frazer is mentioned by Verrips, who noted that scatological practices reveal the "human ability to indulge in all kinds of humiliation and uninhibited aggression, violence, and cruelty toward fellow human beings (as well as other living creatures), in short to waste lives" (Verrips 2017, p. 41).

Cohen Vaxberg sacrifices or destroys a high or refined aesthetic, or an aesthetic that conveys its message in a restrained manner. She prefers to reach a larger audience, to convey her message in a way she believes is effective, to create the aesthetic through the political, and to use art to express clear activism.[19] In this context, it is possible to argue that her work is characterized by what Foster has termed the "pursuit of the referent" or "the politically real, the socially real", a method of working devoted to illuminating "signs of oppression" (Foster et al. 1993, p. 15). Cohen Vaxberg presents herself in the video *Vote Glycerin Suppository* as someone who allows the filth—which hides within Israeli society and drives it—to be released, to come out, to be exposed to sunlight, which, as we know, is the best disinfectant. In comparing herself to the glycerin wick that releases the excrement, she exposes the stereotypical, ethnological, racial, and gender perceptions that police Israeli society, as well as the cultural stereotypes, in order to challenge the existing social order.

The scatological abject art provides "an oppositional practice rather than an ontology", (Taylor 1993, pp. 59–60) presenting a materiality that expresses physicality, sensuality, and difference, while opposing social oppression. According to Simon Taylor, such art is directed against such generalizing and unifying concepts as identity, system, and order (Taylor 1993, pp. 159–60), and the excrement is used to challenge homogeneity and lead to heterology, which Bataille calls "the science of what is completely other", and to scatology, which he calls "the science of excrement" ("science de l'ordure")—a science that, according to him, is the material complement of abstract heterology. (Bataille 1985, p. 102) The materiality that pervades Cohen Vaxberg's works can be understood through this concept of radical "base materialism", which Bataille refers to as "radical materialism". Unlike other types of materialism, "radical materialism" does not provide idealistic meanings that differentiate between high and low or clean and dirty. "Low materialism" (base materialism) seeks to dismantle any systematic basis; it cannot be reduced to a form or order, it cannot be mapped into scientific systems, and it cannot be regulated within a political order. According to Noys, Bataille considers materiality to be an unstable difference, something that cannot achieve logical stability; base materialism is a dynamic occurrence that acts like a virus or bacteria, an infectious thought. (Noys 1998, p. 503) Polluting, profane, anal, and scatological art can be explained by base materialism.

In his book *History of Shit*, Dominique Laporte writes: "We dare not speak about shit. But, since the beginning of time, no other subject—not even sex—has caused us to speak so much. The unnameable is enfolded by strange rumor, which combines the most immaculate silence with the most prolix chatter" (Laporte 2000, p. 112). The call for cleanliness, according to Laporte, is motivated by a fantasy about purity—cleanliness—a fantasy that ultimately clashes with the contamination of the real. He perceives the political demand for cleanliness as echoing the three cornerstones of culture according to Freud— "Cleanliness, order, and beauty", and he adds: "the more it institutionalizes Freud's triad, the more totalitarian the state becomes" (Laporte 2000, p. 56).

**Funding:** This research received no funding.

**Institutional Review Board Statement:** Not applicable.

**Informed Consent Statement:** Not applicable.

**Data Availability Statement:** Not applicable.

**Conflicts of Interest:** The author declares no conflict of interest.

## Notes

1    According to Jessica Marian, "Glas is styled around the "decapitating" of metalanguage through a rhythmic "plunging" and "extracting" of the remaindered or excluded element in and out of the text." Derrida's thorough discussion in his book *Glas* ultimately challenges the perception of a text as an object with a final and conclusive position. (Marian 2015) Derrida, as well as Chakraborty's interpretation of his writings, pave the way for us to comprehend the significance of the body, the Other, the foreign, the surplus, the despised, and the supposedly filthy, the feces.

2    In addition to the combined version, a series of individual videos were also released in which Cohen Vaxberg defecates on the flags of various nations while the national anthem of each nation is playing. (See Shit on the flags of Israel haters 2014).

3    "The Flag, the Emblem and the National Anthem Law, 1949," https://www.nevo.co.il/law_html/law01/067_001.htm (accessed on 27 May 2023). (In Hebrew)

4    As part of the investigation, Cohen Vaxberg was also asked about the video *Vote Glycerin Suppository* (see Fishbein 2015).

5    In 2010, Israeli flags stuck in dog excrement were reported from the streets of Tel Aviv (see Tsofant 2019; "The mountain of shit"). This was done again in 2017 in Rishon Le'Zion (see "Israeli flag stuck on excrement on Rishon Lezion Street," https://www.youtube.com/watch?v=6GRHjjiHNMU&feature=youtu.be, accessed on 20 March 2022). In this video, the documentarian insults an imaginary "leftist" whom she assumes is responsible for the incident.

6    See responses 621, 612, 610, and 607, respectively, along with hundreds of other similar responses to Hila Kobo's article (Kobo 2016).

7    Some of the responses to the work *Poop instead of blood* and the responses to the article "Natali Cohen Vaxberg: I really have no regrets", reproduce the scatological nature of the work but add a distinctly violent tone to it, for example, a video that appears to be a digital rendering of a picture of a man kneeling on a picture taken from the video *Poop instead of blood*; see https://rotter.net/forum/scoops1/152667.shtml, response 12 (accessed on 1 April 2019). The thread is currently unavailable.

8    Many examples can be found in responses to articles that dealt with *Poop instead of blood* on the Internet, for example in the discussion on the "Rotter" website following the article by Hila Kobo, "The artist who defecated on the flag." Thus, for example, the title of response #28, written by someone who calls himself a "publicist", is as follows: "Self-expression??? Every baby who hasn't yet grown teeth expresses himself in a similar way ugh", and in the body of the comment it is stated: "She needs hospitalization, the moron. And I am not a follower of the flag. I have not waved the Israeli flag in my life, and I will never do it either"; see "Rotter", thread no. 280609, https://rotter.net/forum/scoops1/280609.shtml#28 (accessed on 21 March 2020). [Subsequently, the thread on the Rotter website has been removed.]

9    This is not to say that Cohen Vaxberg, either in the framework of the performing arts and theater or in the area of plastic art, was the first to incorporate scatological or abject elements in Israeli art.

10    Individuals or groups of people who are marginalized, excluded, and unable to fully engage in society are referred to as "Wasted Lives" in Zygmunt Bauman's work. In the face of development and economic growth, national politics, and globalization, certain people are characterized as a "waste" in modernity. (Bauman, *Wasted Lives*) In the preface to another of Bauman's books, Yehouda Shenhav states: "The people of the sanitation department, the garbage collectors, are the real agents of modernity, says Bauman." "Day after day, they confirm the distinction between dirt and cleanliness, desirable and unpleasant, accepted and rejected, what remains inside and what remains outside, normalcy and pathology, health and disease/illness. When the waste is collected, the border's path is always decided anew, and this path specifies what waste is." (Shenhav 2007).
Susan Signe Morrison's "The Literature of Waste: Material Ecopoetics and Ethical Matter" is another important literary work worth addressing. In it, she explores instances of both tangible and figurative waste in Western literature. (Morrison 2015).

11    Kristeva's theory has received much criticism over the years, mainly against the background of her claim regarding the rejection of the maternal, the background of her universalist discussion of the abject, and her position on the question of the reification of motherhood. (See Butler 1993, *Bodies That Matter*; Grosz 1994, pp. 187–210; Tyler 2009, 2013 "Against Abjection" and *Revolting Subjects*; Barkardóttir 2016).

12    Bronz also presented the show at the Clipa Theater. The full name of the performance: *Love the Juice: The Show of Squeezing the National Vision into an Orange*; see https://www.clipa.co.il/show/love-the-juice-he/ (accessed on 20 March 2022). At the "Haaretz Culture Conference" Bronz presented only part of the show.

13    An affinity can be found between the show *Love the Juice* and Francesco Clemente's work *La Mia ginnastica* (1982) and his works in the 1990s, which also engaged with the pre-symbolic stages of the present (see Szulakowska 2016).

14    This is the reason for the centrality of art dealing with the abject in feminist contexts, since the definition of the abject is sometimes based on a patriarchal social order; in the 1980s and 1990s, many artists worked within this conceptual world, for example, Cindy Sherman, Louise Bourgeois, Helen Chadwick, Paul McCarthy, Gilbert and George, Robert Gober, Carolee Schneemann, Kiki Smith, Sarah Lucas and Jake and Dinos Chapman, Kiki Smith and Sarah Lucas.

[15]  Fascism is not necessarily associated with a religious power base; however, in Israel, the combining of a religious concept with a hierarchical ethnic concept may lead to a paradoxical view of the word "fascism" as a signifier of the positive (see Azoulai 2019).

[16]  See "Without a free culture there is no democracy", website of the Association for Civil Rights in Israel, 22 October 2018 (accessed on 10 January 2023): https://www.acri.org.il/single-post/85?gclid=EAIaIQobChMI_ez2vuDJ5wIVhbTtCh003gJbEAAYASAAEgLUe_D_BwE.

[17]  Cohen Vaxberg (2014b), from *Response to Arrest (The Poop Affair)*.

[18]  Dr. Heinz Thilo, a member of the SS who worked in the Auschwitz concentration camp, named the extermination facility "Anus Mundi" (see Kępiński 2018).

[19]  This claim is primarily based on Foster's position in the discussion, which dealt with the transition in the exhibition presented at the Whitney Museum in New York in 1993, from dealing with the politics of the signifier to dealing with the politics of the signified, as well as on a suggestion that arose in the discussion to balance the commitment to deal with the signifier and the effectiveness of the artistic action, which seeks to reach broad audiences (see Foster et al. 1993, p. 14).

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
