# Peer review of "Performing Feces in Contemporary Video and Performance Art in Israel"

_arts_

Round 1
Reviewer 1 Report
Overall, a compelling and enlightening reading of a number of key - and artistically varied - critiques of current Israeli-nationalist-Zionist discourses. Individual readings are well contextualized and documented as well as lucidly argued, and the expected "big guns" in 20th- and 21st-century analysis of scatological rhetoric and representation are judiciously and effectively cited. I happily confess that after wading through what seems to me to be an unnecessarily jargon-filled opening, I couldn't put it down. And when I reached the concluding sentence, my first thought was the hope that this was just the opening salvo in a longer study. The appropriate contribution to the proposed multivalent collective volume. About that opening... I cannot help but think it could be made more reader-friendly, although I hasten to add this is minor quibble from a 'simple' reader made grumpy by the assumption we're familiar enough with specific Derrida and Spivak works to absorb with ease the opening of paragraph 2. As I soldiered on into paragraph 3, however, and the sharp individual readings, I found myself on firmer footing. I should also hasten to admit that I had passing familiarity with only one of the works considered - Rosen's "The Dust Channel" - and only what knowledge of contemporary Israeli society and politics I glean from center & left-of-center American journalism. The average Arts reader is likely more sophisticated. I am, after all, an early modernist Europeanist. This article stimulates me to read more, both more broadly and deeply, so for that alone I congratulate its author and warmly endorse its publication.
Author Response
In response to the first review, I have now added a footnote that – to the extent that this form of article allows – slightly explains Derrida's Glas; and the reference list now includes the article cited in this comment. The first and second paragraphs simply serve as a (ethical) foretaste for the discussion that follows, as the reviewer her/himself has noted.
Reviewer 2 Report
Highest marks all around with minor revisions:
I would mention Zygmunt Bauman's Wasted Lives and Susan Signe Morrison's The Literature of Waste as key texts dealing with waste/metaphor (mention on page 1-2 and 11)
correct: "Bronze" sometimes spelled "Bronz"
pg. 11 "Buter" should be "Butler"
Fascinating!
Author Response
In response to the second review:
1. I have unified the spelling of the artist's name – Ariel Bronz and not Ariel Bronze
2. I have corrected the typo on page 11 – "Butler" and not "Buter"
3. At the reviewer's recommendation, I have added in a footnote a reference to the books by Zygmunt Bauman and Susan Signe Morrison. These publications are now also added to the Reference list at the end of the article.